# RNAflow: An Effective and Simple RNA-Seq Differential Gene Expression Pipeline Using Nextflow

**DOI:** 10.3390/genes11121487

**Published:** 2020-12-10

**Authors:** Marie Lataretu, Martin Hölzer

**Affiliations:** 1RNA Bioinformatics and High-Throughput Analysis, Friedrich Schiller University Jena, Leutragraben 1, 07743 Jena, Germany; marie.lataretu@uni-jena.de; 2Methodology and Research Infrastructure, MF1 Bioinformatics, Robert Koch Institute, Nordufer 20, 13353 Berlin, Germany

**Keywords:** RNA-Seq, workflow, Nextflow pipeline, differential gene expression analysis

## Abstract

RNA-Seq enables the identification and quantification of RNA molecules, often with the aim of detecting differentially expressed genes (DEGs). Although RNA-Seq evolved into a standard technique, there is no universal gold standard for these data’s computational analysis. On top of that, previous studies proved the irreproducibility of RNA-Seq studies. Here, we present a portable, scalable, and parallelizable Nextflow RNA-Seq pipeline to detect DEGs, which assures a high level of reproducibility. The pipeline automatically takes care of common pitfalls, such as ribosomal RNA removal and low abundance gene filtering. Apart from various visualizations for the DEG results, we incorporated downstream pathway analysis for common species as *Homo sapiens* and *Mus musculus*. We evaluated the DEG detection functionality while using qRT-PCR data serving as a reference and observed a very high correlation of the logarithmized gene expression fold changes.

## 1. Introduction

More than a decade ago, the possibility to sequence the transcriptome (RNA-Seq) of any given species, starting from small bacteria, such as *Helicobacter pylori* [1], opened up completely new ways to analyze gene expression and obtain never before possible insights into regulatory mechanisms on an incredibly large scale. Starting from short, single-end reads and few samples per study, we now have access to continuously growing and large transcriptomic data sets of high sequencing quality [2]. A standard and widely distributed computational task comprises the processing of RNA-Seq data to identify differentially expressed features, mainly genes (DEGs), between varying conditions, such as different tissue types or control vs. treated samples. Although being frequently performed nowadays, setting up the necessary computational steps to perform an RNA-Seq-based DEG study is still time-consuming and prone to errors, which ultimately leads to results that are difficult to reproduce [3].

However, why are, even after a decade of RNA-Seq tool development and pipeline construction, such studies still difficult to run and reproduce? The main reason is that, as in other bioinformatic areas, RNA-Seq computational studies are complex. Dozens of dependencies for different tools performing sub-tasks, such as quality control, trimming, mapping, and counting, need to be considered and appropriately handled. In addition, the experimental nature of academic software tends to be difficult to install, configure, and deploy. Finally, calculations need to run on heterogeneous execution platforms and system architectures, which range from a small laptop to a high-performance cluster or the cloud. Thus, with increasing access to high-performance computers and cloud resources, the scalability of bioinformatics pipelines becomes increasingly important.

An incredible number of differential expression pipelines were developed in the past because of the wide range of applications that RNA-Seq has [4,5,6,7,8,9,10] and, almost certainly, there is an even larger number of RNA-Seq pipelines developed in parallel in many research groups that will never leave their domestic computing environment. Thus, we decided to develop yet another RNA-Seq pipeline to detect DEGs; however, with the goals to be (1) easy to install, execute, and reproduce, (2) easy to maintain and extend, and (3) to provide reasonable results with a minimal set of input parameters also for non-experts. We achieve these goals by using a state-of-the-art workflow management system and packaging and container systems for each tool included in the pipeline.

## 2. Materials and Methods

### 2.1. Implementation

Our pipeline is implemented in a reactive workflow manager Nextflow v20.10.0 [11], which makes it portable, scalable, parallelizable, and ensures a high level of reproducibility. Furthermore, our implementation allows for selectively re-executing parts of the workflow in the case of additional inputs or configuration changes and to resume execution at the point where the workflow was stopped; this especially important in large-scale and time-consuming RNA-Seq studies. Additionally, time-consuming steps, such as indexing of a reference sequence, are only performed once when executing the workflow multiple times using the same working directory. Each step is encapsulated in a module exploiting dependency management via Conda (https://docs.conda.io) or container usage via Docker [12] or Singularity [13] for an easy installation and execution of the pipeline. Only Nextflow and Conda or Docker or Singularity need to be installed in order to run the pipeline. Currently, the optional transcriptome assembly steps only work with Docker or Singularity due to deprecated BioConda environments for certain tools. We provide configurations for local execution and the Slurm and LSF workload managers. Because Nextflow offers out-of-the-box execution for many schedulers and cloud computing platforms, it can be easily customized to meet individual needs. A recent study compared the currently most commonly used workflow management systems (Snakemake, Nextflow, CWL) with a focus on rapid prototyping for the analysis of ribosome profiling data and, finally, chose Nextflow, which best met the authors’ requirements [14].

The pipeline that is presented here covers all necessary steps for a frequently performed DEG analysis by incorporating a selection of widely used and well established, but yet up-to-date, tools (Figure 1). In the past, we have performed various RNA-Seq expression studies and based our choice of tools on this experience. Therefore, for each step (such as trimming, mapping, and counting), we have selected tools that are highly cited, maintained, usable, and perform well in recent evaluation studies if available. We decided to use an alignment-based mapper, so that the user can actually inspect expression patterns and read mappings subsequently with a genome browser. Still, pseudo-alignment or alignment-free quantification tools, such as kallisto [15] and Sailfish [16], can serve as time- and resource-saving alternatives, and can be considered in a future extension. However, RNA-Seq pipelines that use pseudo-alignment methods are ranked low regarding accuracy [17].

Most importantly, we automatically deal with common pitfalls in DEG detection with RNA-Seq data: ribosomal RNA (rRNA) is removed before mapping, the annotation is filtered for features of interest (genes and pseudogenes), low abundance features are filtered out (TPM filter), and quantification is based on uniquely mapped reads in order to ensure the most accurate result while using default parameters as possible. Although most RNA-Seq protocols include the depletion of rRNA before sequencing, moderate up to high amounts of rRNA can still be found in samples [18,19]. In particular, this is true for non-model organisms where library preparation kits are not optimized [20]. Therefore, we have decided to perform an rRNA cleaning step by default, which can be deactivated by the user. By providing such a default behavior of the pipeline, we allow non-experts to run a DEG analysis with almost no configuration effort to obtain robust results, while experienced users can still customize the pipeline, e.g., by also investigating DEGs that are based on multi-mapped reads. Only a basic understanding of the command line is required to install and run RNAflow. Nextflow and the containerization software can be run on Linux, Mac OS, and Windows supported through WSL2. CPU usage can be adjusted freely and it is automatically restricted to the number of available cores. The minimum RAM requirement heavily depends on the input data. However, in our experience, 8 GB RAM is sufficient for running a human-size RNA-Seq study.

#### 2.1.1. Input

Only two parameters are required in any case: the RNA-Seq reads (single or paired-end) and a reference genome with a matching annotation (GTF format). File paths to the RNA-Seq read files and meta-data are provided via a comma-separated (CSV) table, where each line annotates one sample with a sample name, the path to the FASTQ file, in the case of paired-end data to both files, and the condition (such as ‘control’ or ‘treated’). The read files need to be in FASTQ format and they can be gzipped. For the genome and annotation, the user can make use of an automated download when working with data from *Homo sapiens*, *Mus musculus*, *Mesocricetus auratus*, or *E. coli*, just by specifying ‘--species’ with a three-letter shortcut (‘hsa’, ‘mmu’, ‘mau’, and ‘eco’). Otherwise, CSV files pointing to the genome(s) in FASTA format and matching annotation(s) in GTF format have to be provided. The two input options can also be combined, which is particularly useful, for example, when investigating a viral infection in human cells. In such cases, all provided genomes and annotations are concatenated for further analysis. The pipeline was primarily built to work on Ensembl [21] annotations, but can also handle other annotations, which, however, might lead to reduced features in the final output. For example, we extract certain features from Ensembl annotations, such as the *gene_biotype*, in order to display this information in the final output tables. By default, a DEG analysis with all possible pairwise comparisons in one direction is performed based on the user-defined conditions in the initial CSV file. While using the ‘--deg’ parameter, this behavior can be configured and the user can specify which comparisons should be performed by providing an additional CSV file. With the ‘--assembly’ parameter, one can switch from DEG analysis to transcriptome assembly, where the genome(s) and annotation(s) are used for a reference-based transcriptome assembly. For tools that need a database (SortMeRNA [22], BUSCO [23], and dammit (http://www.camillescott.org/dammit)), the external resources are automatically downloaded and stored in a dedicated project-independent folder. Thus, Nextflow is able to check whether the resource is already available and it skips the download step the next time the pipeline is executed. The same procedure is used for the provided auto-download for genomes and annotations. Accordingly, the pipeline can run offline to a certain extent (pathway analyses still require a connection) if all online resources are initially downloaded.

#### 2.1.2. Preprocessing & Mapping

Quality reports of raw reads and preprocessed reads are generated by FastQC v0.11.9 (https://www.bioinformatics.babraham.ac.uk/projects/fastqc/). Quality trimming and adapter clipping are performed by fastp v0.20.0 [24] in order to filter the raw reads with a sliding window approach and cutting options for front and tail ‘-5 -3’, a default window size of 4 (‘-W 4’), and a default mean quality of 20 (‘-M 20’). The minimum length is 15 bp (‘-l 15’) and the maximal number of N base is 5 (‘-n 5’). PloyX trimming in 3’ ends is enabled (‘-x’). The compression level for the compressed output is set to ‘-z 6’, in order to reduce memory usage in return for a longer compression time. All of the parameters for fastp can be changed with ‘--fastp_additional_params’. We chose to use fastp, because it is faster than other FASTQ preprocessing tools [24] while offering automatic adapter detection and many customization options for experienced users. While using the preprocessed reads, ribosomal RNA (rRNA) is removed with SortMeRNA v2.1b [22] when considering all of the available databases for SortMeRNA. This step can be skipped with ‘--skip_sortmerna’. Note that we stick to an older version 2.1b in order to avoid repeated indexing of the databases, which is not available in versions > 4. Besides, there are no algorithmic changes between versions 2.1b and 4. The trimmed and rRNA-cleaned reads are then mapped with HISAT2 v2.1.0 [25]. The output is directly piped into samtools, converted into BAM format, sorted, and then indexed. If the user provides more than one reference genome, all of the genomes are concatenated and one index is built and used for mapping. We have chosen HISAT2 as mapping software, because it has been continuously developed and maintained for several years [26]. In addition, a recent study on *A. thaliana* showed no major differences between RNA-Seq mapping tools in the context of DEG detection [27], which was also in accordance with a large RNA-Seq pipeline evaluation study recently performed by Corchete et al. [17].

#### 2.1.3. Quantification & Differential Gene Expression Analysis

All provided GTF annotation files are concatenated and filtered for gene and pseudogene features. We use featureCounts v2.0.1 [28] in order to quantify the mapped reads based on this processed annotation, chosen as a quantification tool that is highly cited and used in the community. Per default, we count non-multi-mapped reads on the exon level and summarize the counts into meta-features on the gene level. The strand-specific counting is set to un-stranded by default and it can be changed with ‘--strand 1’ to stranded and ‘--strand 2’ to reversely stranded. If the user is unsure about the strandness of the data, then the pipeline can run first in un-stranded mode, then the alignments can be investigated, for example, via IGV [29], and then the pipeline can be resumed (‘-resume’) after the mapping step using the appropriate strand information. We implemented a TPM (Transcript Per Million) normalization and filter step, as proposed in [30]. For each gene, the TPM value is calculated, followed by the calculation of the mean TPM in each condition. A gene needs to have at least one mean TPM greater than the threshold (default 1) to be considered in the further pipeline. The user can adjust the threshold with ‘--tpm’. All of the results so far are summarized in a MultiQC v1.9 [31] report. The filtered count tables are passed to the DESeq2 module v1.28.0 [32] as the to date most cited and still maintained tool for DEG calling. We apply DESeq2 normalization, as well as different transformations (regularized log (rlog), variance stabilizing, and log2(n+1) transformation). We generate PCA plots of all samples on all types of transformed counts on the 50, 100, and 500 genes with the highest variance and heatmaps on all types of transformed counts on the 50 and 100 genes with the highest counts and most variable genes. Next, we perform pairwise comparisons, per default all possible comparisons in one direction, and shrink the resulting logarithmized fold changes (logFCs) with the lfcShrink and apeglm [33] method (see DESeq2 vignette for details) comparison-wise. The user can specify specific pairwise comparisons with ‘--deg’. The result tables are filtered by adjusted *p*-value < 0.05 and adjusted *p*-value < 0.01. The unfiltered and filtered tables are available in CSV and Excel format, and as searchable and sortable HTML documents, including expression box plots and further information for each gene that is produced by the ReportingTools R package [34]. Furthermore, the results are visualized with volcano (that is generated with EnhancedVolcano, https://github.com/kevinblighe/EnhancedVolcano), MA and PCA plots, sample-to-sample heatmaps, heatmaps on genes with highest counts, on genes with highest logFCs, and on DEGs with the highest logFCs. Many of these plots are also summarized in an HTML regionReport [35], which allows for a first exploration of the DESeq2 results. For some supported species, the pipeline performs an integrated downstream pathway analysis: WebGestalt [36] runs a gene set enrichment analysis (GSEA) for *Homo sapiens* and *Mus musculus*; and, the piano R package [37] performs a GSEA with different settings and consensus scoring for *Homo sapiens*, *Mus musculus*, and *Mesocricetus auratus*. We aim to extend the list of supported species in the future and on-demand.

#### 2.1.4. Transcriptome Assembly

In the absence of an appropriate reference genome, the user might want to first generate a *de novo* transcriptome assembly. Although not the initial focus of our pipeline, we allow, via ‘--assembly’, to switch from DEG analysis to basic transcriptome assembly (Figure 1). Following a recent comparison study [38], we chose Trinity v2.11.0 [39] in order to assemble the rRNA-free or trimmed reads *de novo* accompanied by a reference-based assembly generated by StringTie2 v2.1.2 [40]. Both of the transcriptome assemblies are functionally annotated with dammit v1.2 (http://www.camillescott.org/dammit). Dammit unites different databases for annotation: Pfam-A, Rfam, OrthoDB, and BUSCO. Besides a pure annotation purpose, the results that are provided by BUSCO v3.0.2 [23] can be used as a common metric in order to evaluate the quality and completeness of both transcriptomes that are based on expected gene content premised on evolutionary principles.

#### 2.1.5. Output

For each step of the pipeline, all of the essential results, e.g., trimmed read files from fastp and BAM files from HISAT2, are easily accessible and they can be used for further investigations outside of the pipeline. The plots are provided as PDF files, tables as CSV, and Excel files. A summary or report information are generally accessible through the MultiQC report.

### 2.2. DEG Validation of RNA-Seq with Publicly Available qRT-PCR Data

In order to validate our differential expression analysis procedure, we compared our results with publicly available qRT-PCR data considering these data as the gold standard. We adopted, like Costa-Silva et al. [41], a real RNA-Seq and qRT-PCR dataset, which is part of the Microarray Quality Control (MAQC) project and covers expression data from two conditions that we compare: Stratagene Universal Human Reference RNA (UHR) and Ambion Human Brain Reference RNA (Brain). The UHR data are composed of total RNA from 10 human cell lines. As part of the MAQC project, these are well-characterized datasets, in which, inter alia, over a thousand genes have been examined with qRT-PCR [42,43].

The single-end short reads are available under the NCBI Short-Read Archive (SRA) accessions SRR037445-SRR037451 (UHR, seven biological replicates) and SRR035678, SRR037439-SRR37444 (Brain, seven biological replicates) [44]. qRT-PCR data are available under Gene Expression Omnibus (GEO) accession GSE5350, platform GPL4097, sample accessions: GSM129638-GSM129641 (UHR), and GSM129642-GSM129645 (Brain) [42].

Short-read RNA-Seq data were analyzed by our pipeline (branch: master, commit: b3cf528f3142a2890287ac8445e7ed44ae187331, and Ensembl release 98) on a high performance cluster running the Slurm workload manager. All of the parameters were set to default, except ‘--tpm 0’, to skip the TPM filter. The unfiltered DESeq2 result table (available on the Open Science Framework (OSF) doi.org/10.17605/OSF.IO/SZEFK) was picked for the comparison.

qRT-PCR data were analyzed by adopting an R script (available on OSF doi.org/10.17605/OSF.IO/SZEFK) retrieved from https://www.ncbi.nlm.nih.gov/geo/geo2r/ with default parameters and differential expression analysis was done with limma v3.42.0 [45]. All of the genes with duplicated gene symbols were excluded, in order to ensure a one-to-one mapping between qRT-PCR and RNA-Seq results.

Both unfiltered result tables were merged by gene symbol, so that each gene of the qRT-PCR dataset could be compared with the corresponding result of the RNA-Seq analysis. We considered genes to be differentially expressed with an adjusted *p*-value ≤ 0.05 of qRT-PCR results. logFC correlation was calculated with the Pearson method and plotted while using ggscatter of the ggpubr R package (https://github.com/kassambara/ggpubr).

### 2.3. Transcriptome Assembly Validation

In order to validate the optional transcriptome assembly functionality that is provided by RNAflow, we selected three RNA-Seq read sets from different species that were recently used in a large *de novo* transcriptome assembly benchmark [38]. We downloaded all of the assemblies and the corresponding raw reads for the *E. coli* (eco), *A. thaliana* (ath), and *M. musculus* (mmu) data sets from OSF (doi.org/10.17605/OSF.IO/5ZDX4) and run RNAflow with the ‘--assembly --skip_sortmerna’ options and otherwise default parameters on each data set. We skipped the rRNA depletion step, because no depletion was performed by Hölzer and Marz [38]. The transcriptome assembly calculations were performed with release v1.1.0 of RNAflow. Finally, we benchmarked and visualized RNAflow’s assembly results that were achieved with Trinity and StringTie2, together with the ten different assemblies from Hölzer and Marz [38] for each data set using BUSCO v3.0.2 and MultiQC v1.9. We selected the following BUSCO databases per data set: eco – enterobacteriales_odb9; ath – embryophyta_odb9; mmu – euarchontoglires_odb9. All of the assemblies calculated with RNAflow are available at doi.org/10.17605/OSF.IO/SZEFK.

## 3. Results and Discussion

### 3.1. The Pipeline Produces Well-Structured Output and Comprehensive Insights into DEGs

We ran our pipeline on a well-described data selection that was obtained from the MAQC project, covering expression data from two conditions: ‘UHR’ and ‘Brain’. We executed the pipeline on all samples and performed a pairwise DEG detection between the two conditions. Here, it was not our focus to biologically interpret this data, but to show and evaluate the functionality and output of the pipeline. The final output directory is well-structured based on each step and tool, see Figure 2, allowing for a straight-forward and clean investigation of all the results. The DESeq2 results can be explored in detail at the OSF repository (doi.org/10.17605/OSF.IO/SZEFK). Figure 3 and Figure 4 show a selection of unmodified output that is generated by the pipeline. The expression patterns behind both conditions are highly diverse, which is reflected in the results and plots, e.g. the heatmap showing strong fold change differences between the two conditions (Figure 3A) and the PCA with principal component one (PC1) already covering 99 % of the variance in the data (Figure 3B). Accordingly, the pipeline detected a large set (n = 20,581, comprising 33.9 % of the input) of significantly (*p*-value ≤ 0.05) differentially expressed genes that result in dense volcano and MA plots (Figure 3C,D). The sample-to-sample heatmap (Figure 3E) shows a clear separation of the two conditions and that biological replicates cluster well, which is an important information for the further analysis and interpretation of such a data set. Based on a large set of DEGs, the automatically performed GSEA reveals affected pathways while using WebGestalt and the piano R package (Figure 4A,B, respectively).

### 3.2. RNA-Seq Results of the Pipeline Correlate Strongly with qRT-PCR Results

Overall, 1044 genes were obtained from the qRT-PCR dataset, of which 1001 have a unique gene symbol. 859 genes have an adjusted *p*-value ≤ 0.05. Figure 5 shows the correlations between the logFCs of the DESeq2 result from RNAflow and the limma result of the qRT-PCR analysis. A very high logFC correlation was obtained with a Pearson correlation coefficient R = 0.95 and *p* < $2.2\times 10^{-16}$. Note that the R and *p*-values do not change when changing the adjusted *p*-value threshold to ≤0.01, ≤0.005, ≤0.001, or no threshold at all (see Figure A1, Figure A2, Figure A3 and Figure A4). This shows that our pipeline is generally capable of detecting differentially expressed genes that are based on the performed steps described in the *implementation* part. However, the qRT-PCR data just cover a fraction of genes that are available from RNA-Seq: here, 20,581 of 60,623 genes have an adjusted *p*-value ≤ 0.05.

### 3.3. Transcriptome Assembly Quality Is Comparable with Previously Reported Results

In a recent *de novo* transcriptome assembly study, Hölzer and Marz [38] compared ten different short-read assembly tools on nine RNA-Seq data sets spanning different kingdoms of life. Here, we selected three of these data sets and applied RNAflow’s assembly functionality. Currently, RNAflow implements two approaches for transcriptome assembly: *de novo* via Trinity and reference-based via StringTie2. Next to a functional annotation of the resulting transcripts via dammit, the pipeline also outputs a transcriptome completeness metric via BUSCO. For each data set, we also downloaded the ten assemblies that were previously calculated by Hölzer and Marz [38] and re-analyzed them together with the assemblies that were generated by RNAflow in order to compare the number of single-copy orthologs found for each assembly (Figure 6).

We see that RNAflow is able to produce a robust transcriptome assembly on different data sets (Figure 6), with BinPacker [46] showing the lowest performance in accordance with Hölzer and Marz [38]. Interestingly, we observed differences in the BUSCO assessment between Trinity and StringTie2, which results in generally more complete single-copy orthologs found via the reference-based approach implemented in RNAflow. In particular, for the *E. coli* data set, StringTie2 outperformed all other *de novo* assembly tools. However, a *de novo* approach is often needed in the absence of an appropriate reference genome and annotation. Note that the light differences observed between RNAflow’s Trinity assemblies and the Trinity assemblies previously produced by Hölzer and Marz [38] can be explained by differences in the read pre-processing steps and the different Trinity versions used (v2.11.0 vs. v2.8.4).

Again, we want to stress that RNAflow was primarily developed with the detection of DEGs in mind and the assembly functionality was implemented lately as an optional feature. Thus, other pipelines, such as the Oyster River Protocol [47], should be preferred if the goal is to calculate a comprehensive assembly, in particular, because *de novo* transcriptome assembly is still a challenging task prone to errors [48,49]. Nevertheless, we show that RNAflow is able to generate a basic assembly with comparably good BUSCO metrics in comparison to other assembly tools (Figure 6) and, thus, can be also used for a fast and simple assessment of differentially expressed transcripts in the future.

### 3.4. Comparison with Other RNA-Seq Pipelines

In recent years, various RNA-Seq pipelines have been published. While some implementation features and tool selections overlap with those provided by RNAflow, reproducible installation and execution, as well as a straight-forward usage, differ considerably, see Table A1. The RNA-Seq pipeline provided by the nf-core community (nf-core/rnaseq [9]) also employs Nexflow as workflow management system, but it does not provide such extensive DEG analysis and visualization as implemented in RNAflow. Both pipelines provide true encapsulated containerization, also for the operating system, to minimize effects of numerical instability. VIPER [4] and RASflow [50] make use of Snakemake [51], another popular bioinformatic workflow management tool. Thus, it is easy to parallelize and use Conda in order to install all of its dependencies. In contrast, hppRNA [8] mainly uses Perl (https://www.perl.org/) scripts to create a Snakemake file, but it lacks a true dependency management system. A graphical user interface comes with TRAPLINE [6] provided within the Galaxy framework [52]. OneStopRNAseq [53] offers a web application that requires a login for full usage. The back-end is implemented within the Snakemake framework. In addition, OneStopRNAseq seems to be limited to six model organisms (see https://mccb.umassmed.edu/OneStopRNAseq/about.php). RCP [10] provides scripts for one specific scheduler, Slurm, and is not easily adaptable. Additionally, RNAsik [7] and Nextpresso [5] only support specific schedulers with a limited range of basic RNA-Seq analyses steps.

Taking all of this together, RNAflow combines the most important features of available pipelines (Table A1) to provide a reproducible and scalable RNA-Seq workflow that focuses on a comprehensive DEG analysis with a minimal set of input parameters. Because of its implementation in the Nextflow framework, RNAflow comes with out-of-the-box support for various batch schedulers (SGE, LSF, SLURM, PBS, and HTCondor) and cloud platforms (Kubernetes, Amazon AWS, and Google Cloud).

## 4. Conclusions

Here, we present RNAflow (publicly available at https://github.com/hoelzer-lab/rnaflow), another RNA-Seq-Pipeline, which, however, was implemented within the workflow management system Nextflow and, therefore, aims at a very easy handling. Thus, the pipeline is portable, scalable, parallelizable, and it ensures a high degree of reproducibility, which are urgently needed properties in computer-aided biology and especially in performing standard tasks, such as an RNA-Seq DEG analysis [54]. Not only do we automatically address common pitfalls in RNA-Seq expression analysis, such as rRNA removal and low abundance feature filtering, but we also offer an in-depth differential expression analysis with DESeq2, in contrast to other Nextflow RNA-Seq pipelines, such as nf-core/rnaseq [9]. In particular, our comprehensive R scripts constructed around DESeq2 as the heart of our pipeline, help users to generate reproducible and rapid results. Thus, the output of the DEG analysis includes various visualizations of the result, such as heatmaps, volcano plots, and PCA plots. In addition, we provide all of the intermediate result files in a well-structured way for further analysis and project-specific assessment. We made sure that the input parameters are as simple as possible, so that the users can focus on their scientific question assisted by the results of our pipeline.

### Future Direction

Recently, Di Tommaso et al. [11] showed that the containerization of operating systems is crucial in minimizing numerical instability and increasing reproducibility. Thus, we already implemented Docker and Singularity support for all of our processes. With full containerization of the pipeline, we also plan to distribute RNAflow in the cloud. However, since container technologies, like Docker and Singularity, are still not available for all users—e.g., due to security standards or slow introduction of new software—we also allow the monitoring of software and database versions, as well as version tracking of the pipeline itself, by direct deployment from GitHub with Nextflow while using Conda environments. In the future, we will provide stable release versions of the pipeline that can be directly pulled via Nextflow in order to allow full reproducibility of the DEG results. We will add further profiles to easily deploy the pipeline in different environments while using Conda and/or Docker/Singularity.

Strand-specific sequencing of the transcriptome enhances the accuracy of the gene quantification [55]. However, the user often does not know exactly if and how the data were sequenced strand specific. By examination of the mapped reads, we plan to automatically estimate strandness to provide a more accurate default setting for featureCounts and reduce another common pitfall in RNA-Seq expression analysis. We also plan to further improve the pipeline by adding specific modules and parameter settings e.g., for the automatic detection and masking of PCR artefacts, as well as support for smallRNA-Seq analyses.

Currently, the DEG detection can be only performed based on a provided reference genome. This behavior excludes less studied species for which no or no suitable reference genome and no annotation are available [56]. By combining the transcriptome assembly with the DEG analysis sub-workflow, we intend to directly analyze differentially expressed transcripts in order to overcome this current limitation of RNAflow.

## Figures and Tables

**Figure 1 genes-11-01487-f001:**
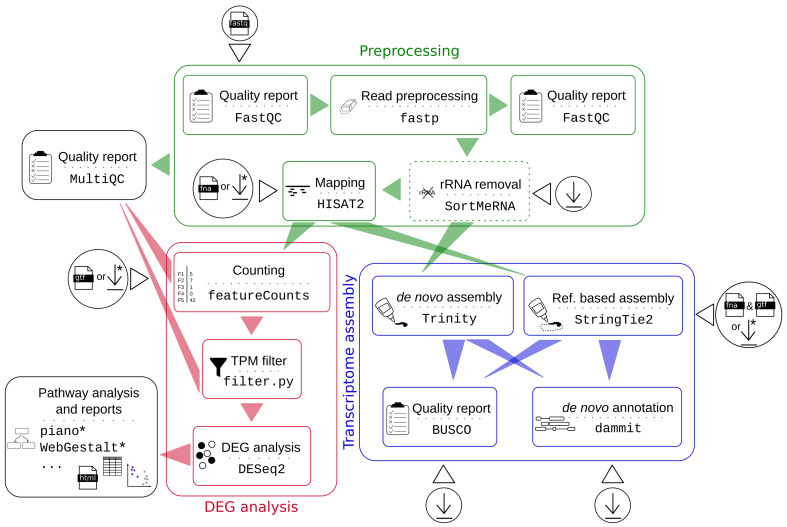
Workflow of the RNA-Seq pipeline. The user can decide after preprocessing to run a differential gene expression analysis or a transcriptome assembly. Circles symbolize input data and download icons symbolize automated download of resources. Steps marked by asterisks are currently only available for some species (see *Implementation*).

**Figure 2 genes-11-01487-f002:**
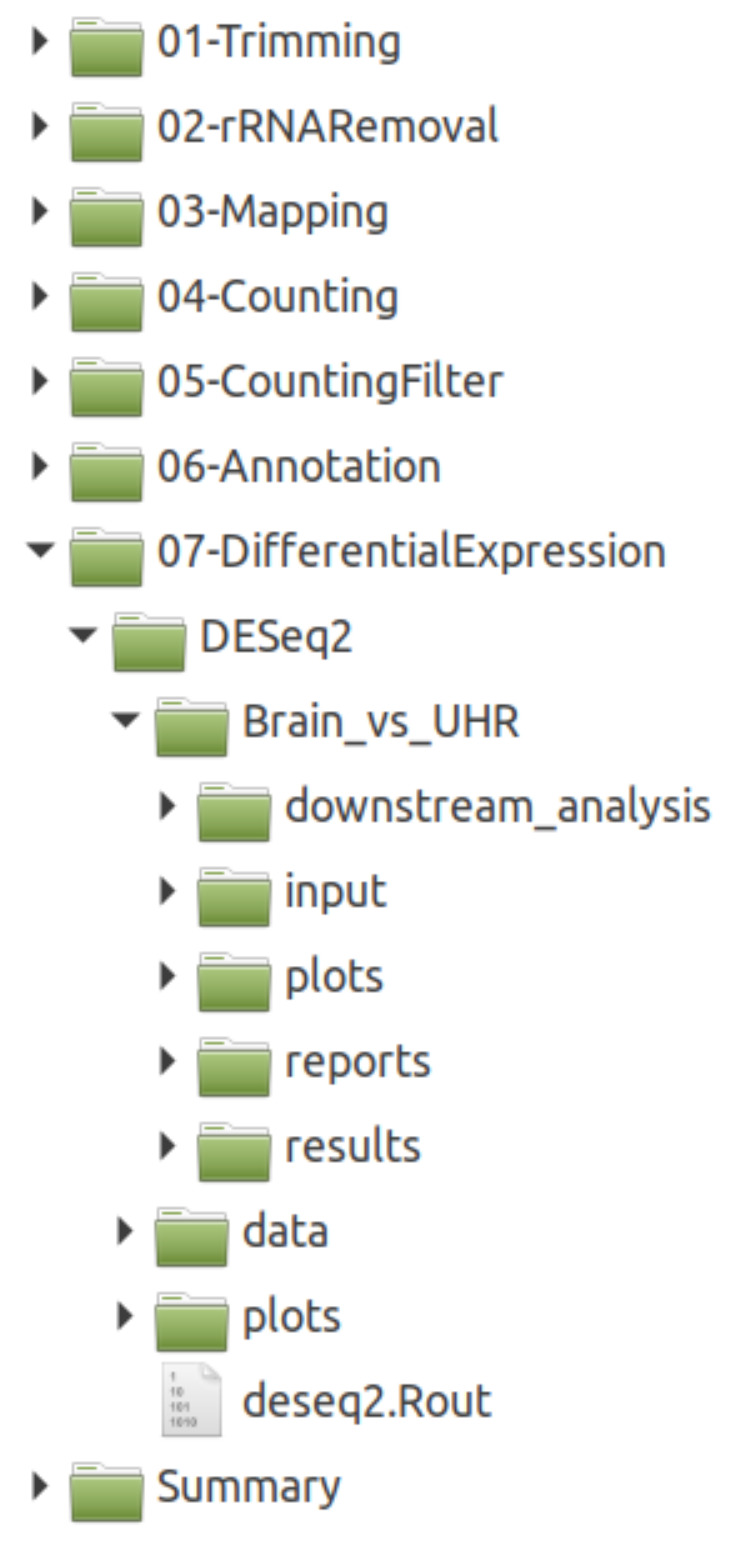
Structure of the result folder. Essential results for each step of the pipeline can be found easily. The DESeq2 output folder comprises general results and statistics for all samples as well as comparison-specific results (in this case, just one comparison was performed: ‘Brain_vs_UHR’).

**Figure 3 genes-11-01487-f003:**
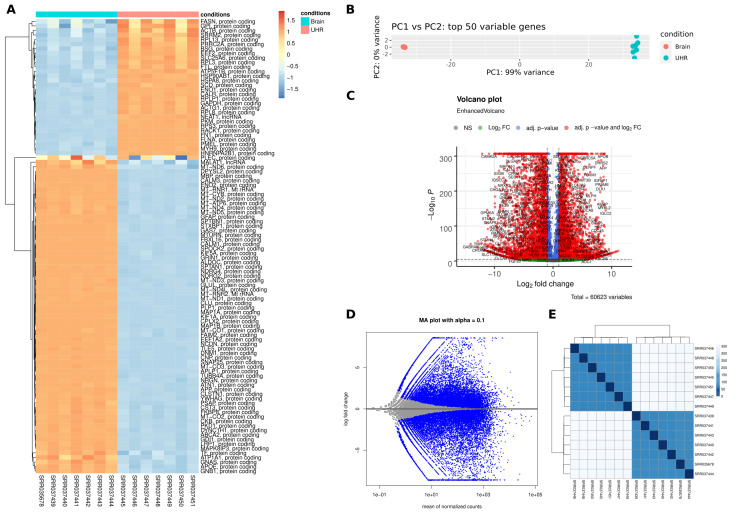
Unmodified visualization plots of a differentially expressed gene (DEG) analysis generated by the pipeline. (**A**) Heatmap on regularized log-transformed counts of the 100 genes with highest gene counts, (**B**) PCA plot on log2(n+1)-transformed counts, (**C**) volcano plot, (**D**) MA-plot, and (**E**) sample-to-sample plot on log-normalized counts.

**Figure 4 genes-11-01487-f004:**
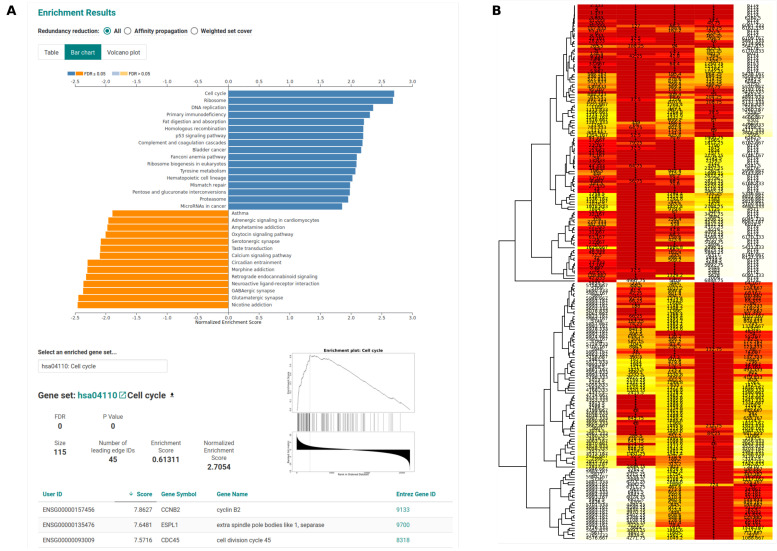
Downstream pathway analysis. (**A**) shows a cutout of the gene set enrichment analysis (GSEA) HTML results generated by WebGestalt, (**B**) shows the consensus scoring of the GSEA result from the piano R package.

**Figure 5 genes-11-01487-f005:**
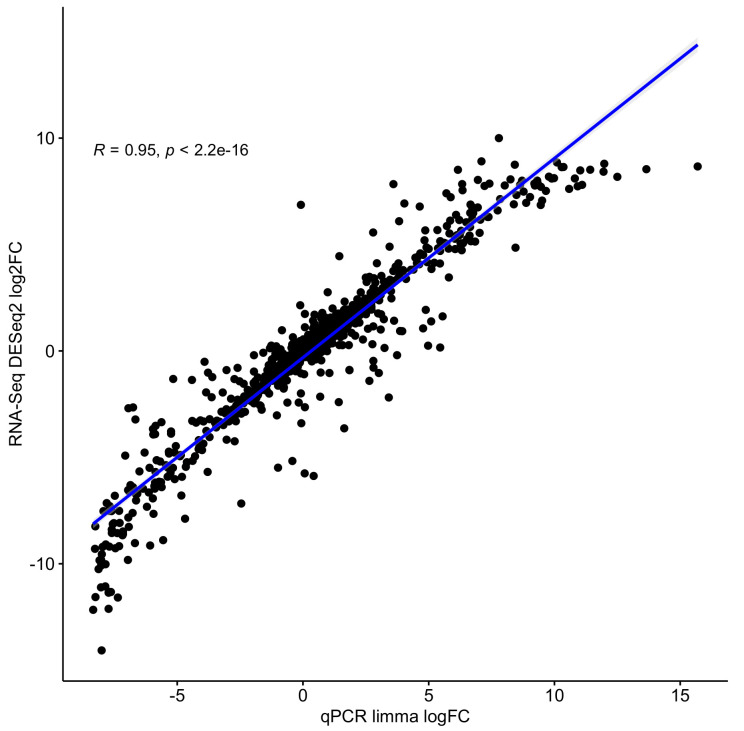
Logarithmized fold change correlation between limma results of qRT-PCR and the pipeline results (DESeq2) of RNA-Seq data filtered by adjusted *p*-value ≤ 0.05 of limma results. The correlation was calculated by the Pearson coefficient.

**Figure 6 genes-11-01487-f006:**
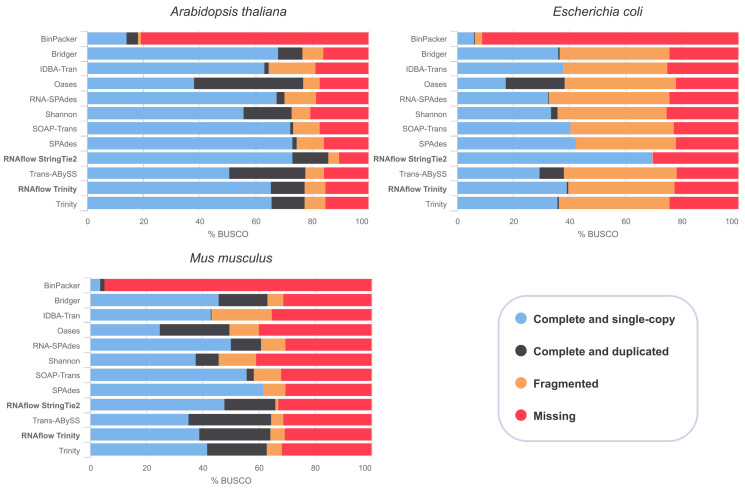
BUSCO assessment. Results of benchmarked universal single-copy orthologs for *E. coli*, *A. thaliana* and *M. musculus* RNA-Seq data sets that were obtained from Hölzer and Marz [38]. RNAflow’s assembly results are indicated in bold. The previously calculated assemblies were obtained from Hölzer and Marz [38].

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
