# Peer review of "RNAflow: An Effective and Simple RNA-Seq Differential Gene Expression Pipeline Using Nextflow"

_genes, 2020, doi:10.3390/genes11121487_

Round 1

Reviewer 1 Report

Leveraging the Nextflow workflow management system, the authors produced a portable, easy-to-install, scalable, and parallelizable RNA-Seq analysis pipeline. The pipeline focuses on differential gene expression analysis, transcriptome assembly, and pathway analysis. In addition, it produces nice plots from the demo run with the example dataset, although the ones in the manuscript are outdated.  While we truly appreciate the efforts that the authors put into this work, we do have some comments as follows.

Major comments:

  1. There are already many pipelines developed in this field and many of them also leveraged various workflow management systems. The authors should briefly list the pros and cons of RN(ext)A-seq against other RNA-seq pipelines, and demonstrate why a new pipeline is necessary.
  2. Suggest authors list the compatible operating systems and the minimum RAM requirement for the pipeline. It also helps if authors can make it clear on the targeted users such as biologists or bioinformaticians.
  3. A rational to remove rRNA before statistical analysis should be added. Most RNA-seq protocols include depletion of ribosomal RNA before RNA sequencing. Which type of RNA-seq protocol will benefit from rRNA removal during analysis?
  4. The authors should provide a table organizing the basic information of the software tools used in the pipeline and also list the supported parameters in the pipeline modules. The required RAM and disk space should be specified. We ran into an out-of-memory issue when running hsa samples on a 16G RAM MacBook Pro. The estimated running time should be documented as well, at least for the example datasets.
  5. Is it correct that the pipeline identified 48680 out of 60623 genes as differentially expressed, which is really high? The normalization in DEseq2 might not be valid with such high number of differentially expressed genes.
  6. The ‘paired’ mode did not work with command ` nextflow run main.nf --cores 4 --reads input.pe.eco.csv --species eco --mode paired --output eco_pe`, which failed in the DESeq2 step. Suggest authors further test the pipeline on different platforms such as Mac, RedHat, and Ubuntu.
  7. Commands to submit jobs to high performance computing environments such as LSF and Slurm systems are not mentioned in Github page
  8. Results are saved in the tmp folder by default and only softlinks are in the results folder, which makes it hard to save results and have ‘reproducibility’.
  9. The commands executed inside the workflow should be saved in the results folder to increase transparency. Log files should also be saved in the results folder.
  10. Need to add case studies on the validity of the transcriptome assembly functionality.

Minor comments:

  1. Currently, the name of the workflow, RN(ext)A-seq, is not easy to pronounce or search online.
  2. In terms of portability (easy-to-install), the author did a very good job. However, in Mac OS, wget is not installed by default which may lead to the termination of the workflow. Suggest author mention this in the Github page
  3. As for documentation, the authors should include a brief tutorial on how to install Nextflow, Conda, and Docker on the GitHub page. Also, wget is not installed in Mac by default.
  4. The contrast CSV file needs to be specified in more detail. For example, which group in the contrast CSV file is the control group? How to prepare the CVS file if there is more than one factor with and without interaction? In addition, the "patient" column seems unnecessary.
  5. We noticed that If a second analysis is run under the same master folder, the previous result folder will be overwritten. The tricky part is, if the second analysis fails halfway, only part of the result folder will be updated, which can be risky and confusing. Better warn users ahead of time if the result folder already exists, so that  users can update the –output parameter.
  6. Line 78, 'Else' needs to be changed to “otherwise”
  7. Line 72 'RNA-Seq reads' can't be provided in csv format. Do you mean the meta-data?
  8. In your experience, does soft clipping in HISAT not sufficient and quality trimming and adapter clipping are necessary?
  9. The PCA plot is hard to interpret with no sample level labeling in Figure 3B.
  10. How were Log-normalized counts calculated in Figure 3A? If library size was not used for normalization, this plot will give misleading information.

The legend in Figure 3 E needs further edits. Do you mean "sample-to-sample plot on log normalized

Reviewer 2 Report

  • It can be a useful addition to bulk RNA-seq data analysis.
  • For the sake of understanding in the paper;
    1. justify the use of every tool in the pipeline, e.g. why DEGeq2 was used and not egdeR; likewise discuss the tools in every step.
  • Additional, methods can be integrated into the pipeline, like spicing event identification and/or RNA based structural variants detection.
  • Make a comprehensive documented tutorial/wiki page for better understanding of the steps and parameters.
  • For the users with limited storage and low-end (local) machines or a biologist without commend-line experience, I strongly recommend developing a web-based implementation of the pipeline.

Author Response

We attach all comments and responses to all three reviewers because we cross-reference between them in some cases. 

Reviewer 3 Report

The authors implemented a full bulk RNA-seq analysis computational pipeline including quality filters, downstream differential gene expression and pathway analysis, or transcriptome reconstruction. The pipeline grounds on the Nextflow framework, which provides computational scalability and efficiency. Conservative parameters have been selected as default, yet the software allows customization. 

The authors provided detailed descriptions of their software implementation and one sample analysis. 

The online manual should be improved. In particular:

  1. the installation instructions (or where to find them) for users not familiar with Nextflow and/or Conda (and Docker if necessary) should be reported;
  2. step by step instructions for the test analysis should be improved, f.i. mention to switch to the git commit 
  3. all options should appear also in the manual, not only on the run help
  4. an overview and little explanation of the output would be nice
  5. example input files should be clearly indicated, for instance, the example of the additional CSV file to customize the comparisons in DE analysis

Clarifying the following points would help the reader to evaluate whether the pipeline suits its needs and to understand some pipeline characteristics:

  1. L307 reference 12 is incomplete
  2. L83 please explain better why annotation other than Ensembl “might lead to reduced features in the final output”. Is it referred to some constraint of the pipeline? Or the authors assume Ensembl annotations are the most complete?
  3. L90 “external resources are automatically downloaded”, what if there is no connection? Can the user set local files/databases in the first instance?
  4. L103 can read preprocessing be skipped as for, f.i. SortMeRNA? 
  5. L106 linearly unmapped reads are retained in the BAM, as separate FASTQ files or they are simply discarded? Having those reads available opens the possibility for alternative downstream analysis such as circRNA detection (either by the user or by future extension of RN(ext)A-Seq). This approach is used in circRNA detection pipelines, such as CirComPara (Gaffo et al. 2017).
  6. L112 Did the authors consider removal/marking of PCR and optical duplicates when multimapped reads are included in the non-default read count? 
  7. L112 Does “summarize” mean “sum”? I.e. gene read count is the sum of (non overlapping) exon read counts?
  8. L115 can the user choose an early end point of the workflow to inspect the results, possibly change the parameters, and continue by resuming the run? Or even run 'slices' of the pipeline? This also relates to the claim at L45 “selectively re-execute parts of the workflow in case of additional inputs or configuration changes and to resume execution at the point where the workflow was stopped” and point 5. Another case: building a genome index is time consuming and can be performed once for many runs. Is this step always performed at each run?
  9. L123 the regularized log transform can take a long time for large datasets. How is this issue handled? Please, consider to skip this normalization either by a parameter or with an automatic guess. Otherwise, warn the user.
  10. L124 which normalization is used for PCA and heatmap? rlog? vst? log?
  11. L125 “the 50 and 100 genes with the highest counts”: usually the most variable genes are used to compare sample conditions, not the most expressed. The same apply for PCA. An heatmap of the most expressed (by TPM, not simply by gene read count) genes can be useful, but an additional heatmap of the DEGs could be generated, in addition to the volcano plot.
  12. L126 when >2 conditions, all possible comparisons will increase the number of multiple tests. Is the P-value adjustment considering all the tests at once or is it performed separately for each comparison?
  13. L128 DESeq2 implements an independent filter to optimize the amount of DEGs, which depends on the P-value filter.  Is independent filtering applied? If so, having two different thresholds, are the results re-calculated upon the different threshold setting?
  14. L137 SRR37451 should be SRR037451

Author Response

(The authors gave the same response as above.)

Round 2

Reviewer 1 Report

The authors have addressed many of our key comments. The revised GitHub pages are now easier to read with more detailed instructions.

The followings remain to be addressed.

Major ones:

  1. Q1. There are already many pipelines developed in this field and many of them also leveraged various workflow management systems. The authors should briefly list the pros and cons of RN(ext)A-seq against other RNA-seq pipelines, and demonstrate why a new pipeline is necessary. In Table 1A, authors did not list any RNASeq workflow for comparisons such as the recently published OneStopRNAseq at https://pubmed.ncbi.nlm.nih.gov/33023248/ and RASflow at https://bmcbioinformatics.biomedcentral.com/articles/10.1186/s12859-020-3433-x, 
  2. Q10. A case study of the assembly workflow is still not given. In addition, although the authors mentioned the implementation of BUSCO for assembly quality assessment in the Method section, we did not see the assembly quality assessment in the results section.

Minor ones:

  1. Regarding the documentation of input.csv file in the Github readme file, the column numbers do not match between the header and the content.

Sample,R,Condition,Sourcemock_rep1,/path/to/reads/mock1.fastq.gz,mock,mock_rep2,/path/to/reads/mock2.fastq.gz,mock,mock_rep3,/path/to/reads/mock3.fastq.gz,mock,treated_rep1,/path/to/reads/treat1.fastq.gz,treated,treated_rep2,/path/to/reads/treat2.fastq.gz,treated,treated_rep3,/path/to/reads/treat3.fastq.gz,treated

Reviewer 2 Report

All questions are answered. 

Author Response

Thank you! We are pleased that we could answer all questions satisfactorily.